# Evaluation of the Antileishmanial Activity of Some Benzimidazole Derivatives Using In Vitro and In Silico Techniques

**DOI:** 10.3390/vetsci12060550

**Published:** 2025-06-05

**Authors:** Mustafa Eser, İbrahim Çavuş, Aybüke Züleyha Kaya, Asaf Evrim Evren, Leyla Yurttaş

**Affiliations:** 1Department of Pharmaceutical Microbiology, Faculty of Pharmacy, Anadolu University, Eskişehir 26470, Türkiye; 2Department of Parasitology, Faculty of Medicine, Manisa Celal Bayar University, Manisa 45030, Türkiye; ibrahim.cavus@cbu.edu.tr; 3Department of Pharmaceutical Chemistry, Institute of Graduate Education, Anadolu University, Eskişehir 26470, Türkiye; aybukezuleyhakaya@anadolu.edu.tr; 4Department of Pharmaceutical Chemistry, Faculty of Pharmacy, Anadolu University, Eskişehir 26470, Türkiye; asafevrimevren@anadolu.edu.tr (A.E.E.); lyurttas@anadolu.edu.tr (L.Y.)

**Keywords:** *Leishmania major*, antileismanial activity, benzimidazole derivates, in vitro, in silico, pteridine reductase 1

## Abstract

Leishmaniasis is a neglected tropical disease caused by a protozoan parasite of the genus Leishmania, characterized by its zoonotic nature and widespread global distribution. Pentavalent antimony compounds have historically been the first-line treatment for this disease, which remains a significant global health concern. Additionally, Amphotericin B (AmpB), miltefosine, paromomycin, and pentamidine are commonly used therapeutic agents. However, drawbacks such as toxicity, limited efficacy, safety concerns, high costs, and emerging drug resistance hinder their use. The first step in identifying new drug candidates involves in vitro and in silico studies. This study assessed the antileishmanial efficacy of newly synthesized benzimidazole derivatives.

## 1. Introduction

Leishmaniasis is caused by protozoan parasites belonging to more than 20 species of the genus Leishmania [1]. The World Health Organization (WHO) has classified it as a neglected tropical disease because of its hazard to public health and the lack of effective and affordable treatments [2]. Leishmania parasites are included in the kingdom Protista and the genus Leishmania. Leishmaniasis, previously characterized as a disease common to poor and rural regions, is currently being identified globally in apparently unaffected locations [3]. Transmission occurs both zoonotically (from animals) and anthroponotically (from humans). While transmission via blood transfusion, transplacental routes, and iatrogenic means is uncommon, several cases have been documented [4,5,6]. Leishmania species exhibit a heteroxenous life cycle, requiring two distinct hosts to complete their development: they inhabit mammalian macrophages and the digestive tract of sandflies [7]. Approximately 90 species of sandflies are recognized as vectors of Leishmania parasites. These sandfly species, especially *Phlebotomus* and *Lutzomyia* species, are vectors of infections that can occur in three different types of living organisms [1]. Additionally, some tick species have also been identified as potential vectors of *Leishmania* spp. [8,9]. Clinical manifestations vary depending on the Leishmania species and the host’s immune response. Thus, the forms can be categorized as cutaneous (CL) (localized and disseminated), mucocutaneous (MCL), and visceral or kala-azar (VL) [10]. Visceral Leishmaniasis (VL), which is referred to as kala-azar, is the most hazardous form. *Leishmania* (*Leishmania*) *donovani* and *L.* (L.) *infantum* are the causes of VL [11]. CL is the second most prevalent type, resulting in chronic ulcerative skin lesions covering the body [1]. These types of infections are mainly caused by *L.* (L.) *major*, *L.* (L.) *tropica*, and *L.* (L.) *mexicana* [11]. Recently, strains of *L.* (L.) *donovani* and *L.* (L.) *infantum*, the pathogens responsible for VL, have been confirmed to cause CL in several areas [12,13,14]. MCL is associated with tissue destruction in the oronasal cavity, larynx, and pharynx [1] and is commonly caused by *L.* (*Viannia*) *braziliensis* [11]. Although the typical symptoms of several Leishmaniasis types have been identified, it is essential to acknowledge the prevalence of atypical appearances outside these forms [15]. As of November 2024, the World Health Organization reported that VL is prevalent in 53 countries (66%), and CL is endemic in 56 countries (62%) [16].

Despite the availability of multiple treatment options, the development of a definitive and optimal drug remains a significant challenge. There is a pressing need to develop highly effective therapies with minimal side effects [17]. Pentavalent antimony has been regarded as the primary pharmacological intervention for Leishmaniasis; however, it is suspected to have cardiotoxicity [18], cirrhosis, pancreatic toxicity [19], and the potential for resistance [20]. Consequently, Amphotericin B (AmpB) (along with its lipid formulation) evolved as a second-line treatment. Miltefosine has been utilized in the treatment of VL and CL and has the benefits of being an oral medication with high efficacy and a short therapy duration, although its significant disadvantages include teratogenicity and the potential for drug resistance. Other drugs repurposed for Leishmaniasis treatment include AmpB, miltefosine, paromomycin, and pentamidine. Furthermore, itraconazole has demonstrated antileishmanial properties due to its antifungal effects [21].

Efforts to identify safer and more effective drugs with reduced toxicity and lower resistance potential have intensified. This has encouraged the exploration of novel compounds and alternative therapies [22]. For the development of effective and non-toxic drugs, it is essential to thoroughly understand the mechanisms of action of the parasite and its pharmacokinetics in the host [23]. Phenotypic compound screening, developments in CRISPR/Cas9 technology, and in vivo bioluminescence imaging applications have established a foundation for the evolution of target-based drug discovery [24]. Moreover, rapid progress in computational sciences has revolutionized the drug discovery process, leading to the development of techniques for molecular docking, activity prediction, compound retrosynthesis, and drug formulation [25]. Numerous in silico studies have been conducted to identify new drug candidates in the fight against Leishmaniasis [26,27,28,29,30,31,32,33,34,35,36,37,38].

Unlike mammals, protozoa belonging to the Trypanosomatidae family are auxotrophic for folate and pterins, thus requiring the enzymatic activities of dihydrofolate reductase (DHFR) and pteridine reductase 1 (PTR1) for their survival [39]. The identification of agents capable of inhibiting these essential pathways is of great importance for antiprotozoal drug development [40,41,42]. In particular, PTR1 plays a pivotal role in resistance mechanisms due to its ability to reduce folate derivatives other than biopterins, and its inhibition is considered crucial for fully arresting the metabolic pathway and preventing the development of drug resistance [42]. Indeed, several studies have previously reported that agents targeting PTR1 exhibit notable antiprotozoal activity against Leishmania species [43,44,45]. This therapeutic potential has motivated medicinal chemists to design novel antileishmanial compounds specifically targeting this enzyme [41,43,46].

Literature reviews conducted to identify suitable pharmacophore groups for antileishmanial drug design suggest that compounds containing a benzimidazole scaffold possess considerable potential [33,47,48]. Furthermore, studies have shown that the antiprotozoal effects of benzimidazole derivatives may be attributed to their inhibitory activity against PTR1 [40,41]. It is also noteworthy that the activities of benzimidazoles on folate pathway enzymes such as PTR1 have been associated with their similarity to the core structure of folic acid [42].

Based on this information, the present study was designed to evaluate the inhibitory potential of selected benzimidazole derivatives, previously synthesized by our research group [49] against L. major promastigotes, and to explore their potential molecular interactions with PTR1 using in vitro and in silico approaches.

## 2. Materials and Methods

### 2.1. Chemistry

o-Phenylenediamine and lactic acid were refluxed in 4 N of HCl solution for 8 h. Subsequently, the mixture was poured into ice water and neutralized with ammonia solution. The precipitate was filtered. To synthesize 2-acetylbenzimidazole, 2-(1-hydroxyethyl)benzimidazole was dissolved in acetic acid and heated at 90 °C. A solution of chromium trioxide was slowly added, and the temperature was maintained at 90 °C. After cooling, the mixture was poured into water, and the precipitate was filtered. After that, 2-acetylbenzimidazole and 2-bromo-4′-chloroacetophenone were stirred with potassium carbonate in acetone to obtain **K1**. Compounds **K2**, **K3**, and **K4** were synthesized by 2-(2-acetyl-1H-benzimidazol-1-yl)-1-arylethanone, and ammonium acetate was refluxed in acetic acid. After cooling, the mixture was poured into distilled water, and the precipitate was filtered. (Figure 1) [49].

### 2.2. Cultivations of the L. major Isolates

Protozoan isolates were obtained from the Parasite Bank at the Department of Parasitology, Faculty of Medicine, Manisa Celal Bayar University. The subsequent procedure for acquiring promastigotes was as follows: the major *L.* (L.) isolate was classified as MHOM/TR/2014/CBL-LM. It was preserved in liquid nitrogen and defrosted by placing it in a water bath at 37 °C for two minutes after its removal from preservation. Subsequent to melting, the NNN medium was inoculated and incubated in an incubator at 25 °C. The reproductive condition of the protozoan was assessed by making coverslips and slides twice daily. Following parasite replication, the cells were transferred from the NNN medium to the RPMI-1640 medium (Gibco, Paisley, Scotland, UK) to facilitate further proliferation. Before application, 10% fetal calf serum, 1% penicillin/streptomycin, and 1% gentamicin were added to a widely available commercial medium. Distribution was conducted using 5 milliliters per 25 mL flask, and cultivation was performed by adding 50 μL of the cultured promastigotes. Flasks designated for cultivation were incubated in a 25 °C incubator. Parasite reproduction was observed, and the medium was supplemented every 2–3 days. Subsequently, the number of promastigotes was quantified using a Thoma chamber, yielding a concentration of 10^6^ promastigotes/mL, and a suspension comprising promastigotes was formulated [50].

### 2.3. Determination of In Vitro Antileishmanial Activity

The synthesized compounds were dissolved in DMSO to a final concentration of no more than 0.5% [51]. Based on preliminary studies, the **K1** compound was prepared at concentrations ranging from 0.23 µg/mL to 30 µg/mL, while the **K2**, **K3**, and **K4** compounds were prepared at concentrations ranging from 1.95 µg/mL to 250 µg/mL.

The assay was conducted using 96-well flat-bottom microtiter plates, with three replicate wells per experimental group. The groups included a blank control, parasite control (no treatment), drug control (Amphotericin B), and test groups for compounds **K1**, **K2**, **K3**, and **K4**. Each well was filled with 100 µL of RPMI-1640 medium supplemented with 10% fetal calf serum, 1% penicillin/streptomycin, and 1% gentamicin. For the four test compounds and Amphotericin B, 100 µL of the respective solution was added to the first well of each group. The contents were mixed thoroughly using an automated dispensing pipette, followed by serial dilutions down the respective wells. Promastigote suspensions, previously counted using a Thoma counting chamber, were added at 100 µL per well to all wells except those designated as blanks. No parasites were added to the blank wells. The plates were then covered with lids, sealed with parafilm, and incubated at 25 °C. Following incubation, cell viability was assessed using the CellTiter-Glo^®^ Luminescent Cell Viability Assay Kit (Promega, Madison, WI, USA). This experimental procedure was repeated in three independent experiments conducted at different time points [52,53].

### 2.4. Cytotoxicity Tests

The cytotoxic effects of the benzimidazole derivatives and AmpB were determined using the L929 cell line (American Type Culture Collection, Manassas, VA, USA). In total, 100 µL of the cell suspension (10^5^/mL) was added to each well comprising sterile, flat-bottomed 96-well microplates. The microplates were then incubated at 37 °C in a 5% CO_2_ incubator for 24 h. Cytotoxic activity was evaluated using the same concentrations applied in the antileishmanial activity assessment. For compound **K1**, serial dilutions were prepared in the range of 0.23–30 µg/mL, and for the other compounds, in the range of 1.95–250 µg/mL. These dilutions were added to a second microplate, and 100 µL from each well was transferred to the first plate. The plates were then incubated under the same conditions for an additional 48 h.

Cell viability was determined using the 3-(4.5-dimethylthiazol-2-yl)-2.5-diphenyltetrazolium bromide) (Sigma, Kawasaki, Japan) method. After the completion of the incubation period, all media from the microplates, both containing and lacking the active ingredient, were discarded. Subsequently, 100 μL of MTT suspension was added to each well, and the plates were incubated in a CO_2_ incubator at 37 °C for four hours. At the end of the incubation period, all wells in the microplates were emptied, and 100 μL of pure DMSO (to dissolve the formazan crystals) was pipetted into each well and incubated under the same conditions for 30 min. Absorbance was measured at 570 nm using a spectrophotometer (Biotek 800TS, Winooski, VT, USA). CC_50_ values were determined using GraphPad Prism version 8.4.2. All experiments were performed in triplicate, with three independent trials and two replicates per trial [54].

### 2.5. Selectivity Index

The selectivity index (SI) is a crucial parameter used to evaluate the potential of natural or synthetic compounds, substances, or drugs to selectively inhibit parasite growth, representing the desired therapeutic activity. It is calculated as the ratio of the cytotoxic concentration (CC_50_) required to cause 50% cell death in mammalian cells to the inhibitory concentration (IC_50_) required to kill 50% of the parasites. The selectivity index values of benzimidazole derivatives for *L.* (L.) *major* promastigotes were determined using the following equation: SI = CC_50_/IC_50_ [55].

### 2.6. Molecular Docking and Molecular Dynamics Simulation Studies

Molecular docking studies were performed using an in silico approach as described previously [56]. Crystal data were obtained from the Protein Data Bank server (PDBID: 5L4N) Protein Data Bank. The enzyme crystal was prepared using the Protein Preparation Wizard protocol of Schrödinger Suite 2020, using the Schrödinger Maestro interface for the molecular docking study. The LigPrep module was used to prepare active molecules (**K1** and **K2**) [57].

Docking simulations were conducted following established protocols [58]. The ligand structures were processed using the LigPrep module with the OPLS4 force field. Protonation states were assigned at a physiological pH of 7.4 ± 1.0, and ligands were energy-minimized accordingly. The receptor structure of pteridine reductase 1 (PTR1) was prepared by removing all crystallographic water molecules and cofactors. The protein was then processed using the Protein Preparation Wizard, which included the assignment of proper protonation states, the completion of missing side chains, optimization of hydrogen bonding networks, and minimization of surface charges. Following this, a receptor grid was generated using the Receptor Grid Generation module [59]. Molecular docking was performed using the Extra Precision (XP) protocol. The docking poses were analyzed, and the most plausible binding conformations were selected based on docking scores and consistency with the experimental activity profile. **K1**, identified as the most active compound, was selected for the molecular dynamics simulation (MDS) to evaluate the temporal stability and interaction behavior of the ligand–protein complex. The MDS method, as previously performed by our team, was applied for 100 ns [60,61].

The three-dimensional structure of the best-ranked docking pose of the K1–PTR1 complex was prepared using the System Setup interface. The system was embedded in a POPE lipid bilayer, solvated with TIP3P water molecules, and neutralized by the addition of Na^+^ and Cl^−^ ions. Following energy minimization and system equilibration, the simulation was carried out for 100 nanoseconds at 315 °K under standard conditions.

## 3. Results and Discussions

### 3.1. Antileishmanial Activity

The effects of the synthesized compounds on the viability of *L.* (L.) *major* promastigotes during their reproductive phases are illustrated in Figure 1 and Figure 2, showing the effects of AmpB on promastigotes during the logarithmic growth phase. Figure 1 indicates that **K3** (IC_50_ = 45.11 µg/mL) and **K4** (IC_50_ = 69.19 µg/mL) possessed antileishmanial activity at high concentrations against *L.* (L.) *major*, but **K2** (IC_50_ = 8.89 µg/mL) showed antileishmanial activity at a comparatively lower dosage. **K1** (IC_50_ = 0.6787 µg/mL) was identified as the compound with the lowest concentration among the derivatives (Figure 2). The IC_50_ value of AmpB, which was used as the reference drug in this study, was 0.2742 µg/mL (Figure 2).

In this study, cytotoxicity tests were conducted on these four compounds, and their activities were investigated. The toxicity studies with the L929 cell line obtained the following CC_50_ values: **K1**, 250 µg/mL; **K2**, 63 µg/mL; **K3**, 0.56 µg/mL; and **K4**, 292 µg/mL. The selectivity indices (SI) were calculated as follows: **K1** = 368.25, **K2** = 7.08, **K3** = 0.012, and **K4** = 4.22.

The morphological alterations caused by compound **K1** at various concentrations in *L.* (L.) *major* promastigotes were assessed at the 48 h mark utilizing Giemsa staining (Figure 3). The strains from samples collected from the control well exhibited a specific rosette formation produced by the promastigotes during reproduction (Figure 3a). In samples taken from serial dilutions of **K1** ranging from 0.23 µg/mL to 30 µg/mL, decreases in the number of promastigotes and morphological structural disruptions were observed (Figure 3b–l).

Leishmaniasis is a neglected zoonotic disease and a serious public health issue. Current treatment options face numerous limitations, including severe drug-related side effects [23]. Consequently, further research is essential to identify novel therapeutic candidates. The antileishmanial activity of various compounds bearing benzimidazole moieties has been extensively investigated. In in vitro studies, the efficacy of such compounds against promastigotes is typically classified based on their IC_50_ values. Consequently, they have been categorized as very active (IC_50_ < 10 μg/mL), active (IC_50_ > 10 < 50 μg/mL), moderately active (IC_50_ > 50 < 100 μg/mL), and inactive (IC_50_ > 100 μg/mL) [53,62]. Based on this classification, **K1** (IC_50_ = 0.6787 µg/mL) had an IC_50_ value lower than 1 µg/mL, and **K2** (IC_50_ = 8.89 µg/mL) had an IC_50_ value less than 10 µg/mL, indicating their significant activity against *L.* (L.) *major* promastigotes. **K3** (IC_50_ = 45.11 µg/mL) was found to be active, while **K4** (IC_50_ = 69.19 µg/mL) was moderately active. **K1** was identified as the compound showing the closest antileishmanial effect to the antileishmanial activity demonstrated by AmpB. Hence, when the compounds were evaluated based on their antileishmanial activities, **K1** displayed the highest efficacy. Subsequently, regarding efficacy, **K2**, **K3**, and **K4** were ranked.

Based on cytotoxicity assessments (CC_50_), **K4** exhibited the lowest toxicity, followed by **K1**, **K2**, and **K3**, respectively. The selectivity index (SI) for each compound was calculated as the ratio of CC_50_ to IC_50_ (SI = CC_50_/IC_50_) [60]. A compound is typically considered a promising drug candidate if it exhibits an IC_50_ < 3 µg/mL and an SI > 10 [51]. According to these criteria, **K1** (IC_50_ = 0.6787 µg/mL < 3 µg/mL and SI = 368.25 > 10) was identified as a potential drug candidate among these substances. Although AmpB displayed a lower IC_50_ value, the compound **K1** demonstrated a more favorable selectivity index (SI = 368.25), suggesting its potential as a safer and more effective drug candidate.

### 3.2. Molecular Docking Studies

The two active compounds (**K1** and **K2**) were docked to the active site of PTR1 to determine their binding modes (Figure 4 and Figure 5). The compounds were docked into the 5L4N crystal form of the enzyme. Based on these results, **K1** interacts with Phe-113, Gln-186, Met-183, Asp-181, and NDP-602 via π–π stacking. Similarly to **K1**, **K2** showed π–π stacking interactions with Phe-113, Gln-186, Met-183, Asp-181, and NDP-602.

### 3.3. Molecular Dynamics Simulation Study

Compound **K1**, which showed the highest PTR1 inhibitory activity among the synthesized compounds, was investigated using MDS. MDS is a valuable technique for estimating the time-dependent stability of molecular systems, such as apoproteins, enzyme-ligand, and receptor–ligand complexes. According to the findings (Figure 6A–C), the RMSD values for the protein remained within the range of 1–3 Å throughout the simulation, indicating acceptable structural stability. The RMSF values for rigid secondary structures such as α-helices and β-sheets were below 1 Å, and no drastic fluctuations were observed in loop regions or at residues involved in ligand interactions. The ligand RMSD and radius of gyration (Rg) values also showed deviations within acceptable limits. These results collectively indicate that the complex maintained structural stability during the simulation.

According to the plots in Figure 6D–F and the video, it was observed that compound **K1** formed aromatic hydrogen bonds with Leu226, Gly240, Tyr283, Arg17, Phe113, Tyr191, Asp181, Ser284, Met183, Val228, and Leu188, and also formed π–π stacking with Tyr191. Compound **K1** was contacted via a water-mediated hydrogen bond with Tyr191 (12% and 13%) and a hydrogen bond with Leu226 (16%), Gly240 (13%), Tyr283 (10%), and Arg17 (10%).

Compound **K1** interacted with Phe113 (a hydrophobic interaction), Tyr191 (a water-mediated hydrogen bond, hydrophobic interaction, and hydrogen bond), Leu226 (water-mediated hydrogen bonds, hydrophobic interactions, and hydrogen bonds), and Gly240 (a water-mediated hydrogen bond and hydrogen bond).

Additional interactions included hydrophobic contact with Phe113; water-mediated, hydrophobic, and conventional hydrogen bonding with Tyr191; and water-mediated and conventional hydrogen bonds with Leu226 and Gly240. The interaction between compound **K1** and Phe113 was stable for approximately the first 40 ns, predominantly as a hydrophobic interaction.

## 4. Conclusions

The conclusions of this study demonstrate the efficacy of synthetic benzimidazole derivatives against *L.* (L.) *major*. Based on these findings, we propose that compounds exhibiting antileishmanial activity may serve as potential candidates for antileishmanial drugs, pending further validation of their safety and efficacy through additional in silico, in vitro, and ultimately in vivo studies. Further alterations could augment these compounds’ activities based on their chemical structure. Subsequently, their selectivity towards mammalian cells should be evaluated against parasites. For further studies, the inhibitory effect of **K1** on the PTR1 protein may be evaluated to optimize potential agents. The findings indicated that benzimidazole derivatives display inhibitory activity against *L.* (L.) *major* compared to the pyrazinobenzimidazole ring system. Moreover, the 3-chlorophenyl derivative of the benzimidazole-based compounds was more effective, and its cytotoxic effect on L929 cells was not observed. In conclusion, *N*-alkyl benzimidazole-based compounds exhibited potential inhibitory activity against *L.* (L.) *major* promastigotes.

## Data Availability

No new data were created or analyzed in this study. Data sharing is not applicable to this article.

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
