# Peer review of "Evaluation of the Antileishmanial Activity of Some Benzimidazole Derivatives Using In Vitro and In Silico Techniques"

_vetsci, 2025, doi:10.3390/vetsci12060550_

Round 1

Reviewer 1 Report

Comments and Suggestions for Authors

Mustafa ESERet al. reported Evaluation of the Antileishmanial Activity of Some Benzimid- 2
azole Derivatives Using In Vitro and In Silico Techniques. It is suitable for publication after major revision.

  1. The manuscript presents results that are largely derivative of existing studies on benzimidazole derivatives. The authors need to demonstrate what unique insights their research contributes. A more thorough literature review highlighting the novelty of their compounds and comparing them with previous findings is required.
  2. Introduction is so large. The introduction should clearly delineate the aims and objectives of the study after providing background context, rather than jumping straight into the methods.
  3. The study focuses on in vitro data, the lack of mention of in vivo validation is a significant shortcoming. The authors should discuss the steps required to transition these findings from cell cultures to animal models and the potential challenges involved in this translation.
  4. The manuscript lacks a precise description of the docking protocol used. It is essential to provide detailed methodologies for the preparation of both the receptor (pteridine reductase 1, PTR1) and ligands (the benzimidazole derivatives). This includes steps such as the optimization of the ligand structures, the protonation states considered, and any constraints or parameters employed during the docking process.
  5. The selection of the active site for docking should be justified. Explain how the binding site was determined and whether any specific residues were targeted or if a broader site was used. Discuss if prior knowledge (e.g., crystal structures or previous studies) influenced the selection of the binding site.
  6. The manuscript mentions performing molecular dynamics simulations, it lacks descriptions of the simulation parameters. Important factors such as the length of the simulation, force field used, and how the system was prepared (e.g., solvation, ion addition, energy minimization) should be included.
  7. Additionally, provide justification for the chosen simulation settings and how they are appropriate for assessing the stability of the docking results.
  8. Clarify which scoring function was used to evaluate the binding affinity of the ligands to the target protein. Different scoring functions can yield varied predictions, and this choice should be clearly explained. Furthermore, provide reasonings for selecting a particular scoring method, and discuss any limitations of that method.
  9. There should be a clearer validation strategy for the docking results presented. Discuss whether docking studies were validated against any known inhibitors of PTR1 to assess the reliability of the predictions. Including preliminary results for these known compounds can help substantiate the findings.
  10. It would be beneficial to discuss how the docking results correlate with the experimental in vitro findings. For example, if certain compounds exhibited high binding affinities, did these findings correlate with high efficacy in the cytotoxicity assays? This connection should be explicitly made to reinforce the relevance of the docking studies.
  11. In the context of antileishmanial drug design, discuss how the binding properties of the compounds in relation to PTR1 could relate to potential drug resistance mechanisms. This could involve analyzing whether modifications to the ligands could yield less effective binding due to mutations in the target protein.
Comments on the Quality of English Language

The use of passive voice is prevalent in the manuscript. While passive voice is common in scientific writing, some sections could be strengthened by using active voice to enhance engagement. For example, instead of "the compounds were synthesized," consider "we synthesized the compounds."

There are several grammatical errors and awkward phrasings that need to be corrected.

Author Response

Dear Reviewer,

Thank you very much for your evaluations. According to your evaluation, the activities we have undertaken regarding the changes you requested are detailed in the attachment and in the explanation section. Respectfully.

Reviewer 2 Report

Comments and Suggestions for Authors

The authors focused on Anthelmintic activities of previously synthesized benzimidazole derivatives.

There is no chemistry is this study, as the target compounds were already prpaed and earlier published https://doi.org/10.17776/csj.1392037

There is no rational design for the compounds.

The docking studies are very poor.

Comments on the Quality of English Language

The English could be improved to more clearly express the research.

Author Response

Dear Reviewer,
Thank you very much for your evaluations. According to your evaluation, the activities we have undertaken regarding the changes you requested are detailed in the attachment and in the explanation section. Kind regards.

Reviewer 3 Report

Comments and Suggestions for Authors

The scale shown in the figures (100 µm) should be reviewed. Promastigotes are elongate, slender and measure about 10-12 µm in length (CDC, DPDX).  The magnifications at which microphotographs are observed generally use lowercase letters in the "x" : (x100 magnification).

The lines 65-66: "The primary intermediate hosts are Forcipomyia spp. (Diptera: Ceratopogonidae) " must be reviewed.

The line 180: doses via serial dilutions using a 2% DMSO solution. Can you support with bibliography that this concentration of DMSO does not affect the results?

Line 162: In the methodology I recommend doing the separation of parasite culture and the in vitro evaluation.

I recommend rewriting the in vitro evaluation, as it is confusing, and it is not clear how the compounds and plate were prepared. Also, in some section clarify the reason why different concentrations are used for K1 than for the rest.

Again, in the cytotoxicity test clarify the use of these specific concentrations. Clarify the concentration of DMSO used to dissolve the crystals

As a recommendation, combine the compounds in a single graph to observe the comparison with the control, or at least in a single graph the compound K1 with AmpB

In lines 292-293 it is mentioned that K1 has better activity than AmpB. Based on what? Since IC50 of AmpB is lower than that of K1.

The graphs shown in figure 7 cannot be observed accurately, they are out of focus, I recommend making them larger and thus improving sharpness.

Author Response

Dear Reviewer,
Thank you very much for your evaluations. According to your evaluation, the activities we have undertaken regarding the requested changes are detailed in the attachment. Kind regards.

Reviewer 4 Report

Comments and Suggestions for Authors

Major Concerns:

  • The authors claim that in-house, molecular data banks were screened, and four benzimidazole-based molecules were chosen to evaluate antiprotozoal activity. How were these compounds chosen? The authors fail to provide this critical information in the Methods and Results section.
  • The study is quite preliminary and limited regarding the number of derivatives synthesized, and the authors do not provide a clear sense of the emerging SAR.
  • The rationale for selecting benzimidazole derivatives for testing against Leishmania is also unclear. Is the target known and conserved, or is this just a repurposing opportunity? Further, the authors investigated interactions between these compounds and the active site of pteridine reductase 1(PTR1). What is the rationale for this investigation? The authors can’t randomly pick targets and dock their compounds against their structures.
  • In silico studies, including molecular docking and dynamics simulations, provide limited value in clarifying the potential mechanisms of action for drugs unless backed by experimental results from in vitro work, such as target-based biochemical assays, mutagenesis experiments, and more.
  • Introduction: The authors present an overly detailed review of the lifecycle, global burden, species, treatments, and more in humans while failing to address the significance of these parasites in animals. They need to justify the publication of their manuscript in a Veterinary Sciences journal. Additionally, much of the content is repetitive throughout the introduction (e.g., Lines 114-118). A concise introduction with relevant information sets the stage for a strong manuscript for readers.
  • Figure legends: All figure legends would benefit from more detailed information to ensure that readers can fully comprehend the results without having to refer to the methods or results sections of the manuscript.
  • Methods: Compounds K1 and K2 have the same R group but are different compounds. Do these compounds have a common core structure like the one shared between K2, K3, & K4?
  • Methods: Research has demonstrated that 2% DMSO can significantly reduce cell viability. Therefore, the use of 2% DMSO in this study is not ideal and may contribute to the toxicity and other effects observed with the compounds. It is recommended to use DMSO at a maximum concentration of 0.5%.
  • Methods: The authors employed a luminescence-based viability assay (CellTiter-Glo) to measure the viability of treated promastigotes. However, it is peculiar that they measured the absorbance instead of luminescence for the viability calculations.
  • The results and discussion section is also toned down and would benefit from further synthesis of the discussions comparing these findings to existing knowledge in the field.
  • This in vitro and in silico study does not address the research gap highlighted by the authors in the Introduction section. Further studies are needed to support the claims made by the authors based on their findings

Minor suggestions:

  • Figures 1, 2, and 3 can be combined into a single figure.
  • The figure legend for Figure 4 does not specify what concentrations of K1 were used for the individual images.

Overall, the study in its current form is too preliminary for publication in its current form; therefore, I recommend a more comprehensive study.

Comments on the Quality of English Language

A well-written and grammatically correct manuscript is easy to follow and essential for readers to understand the significance of the research and how the findings presented in this manuscript relate to the current literature. The authors should enhance this aspect.

Author Response

(The authors gave the same response as above.)

Round 2

Reviewer 1 Report

Comments and Suggestions for Authors

accept

Author Response

We thank you for your positive evaluation.

Kind regards.

Reviewer 2 Report

Comments and Suggestions for Authors

The authors' responses are not clear.

Author Response

Dear reviewer,

First of all, it is true that the test compounds in this study are not original compounds. The findings related to the syntheses have been previously published in a chemistry journal. We have clearly stated this information in the relevant section of the manuscript (you can see it in the revised file).  On the other hand, the antilesmanial effects of the tested compounds were demonstrated for the first time in this study and these findings are still important in the context of biological activity. In addition, an in silico approach was also presented in this study regarding the mechanisms of the antilesmanial effects of the test substances. In this context, we believe that our study is scientifically original and valuable.

We also acknowledge your criticism that the hypothesis of our manuscript was not properly stated in the text. Unfortunately, we were not able to express ourselves correctly in the last revision. In this revision, we revised once again the paragraph on why we investigated the antileishmanial activities of our test compounds and tried to be more clear.

Finally, regarding the in-silico studies, we made extensive revisions based on the comments of other reviewers. However, if you have any other specific corrections that you would like us to make, we will gladly try to do so.

It is our hope that you will find our manuscript more understandable and improved in this version. Thank you for your valuable contributions.

Kind regards.

Reviewer 3 Report

Comments and Suggestions for Authors

I appreciate you considering my suggestions.

Author Response

Thank you for your positive approach and valuable contributions.

Kind regards.

Reviewer 4 Report

Comments and Suggestions for Authors

The manuscript has been significantly improved. However, I still fail to understand how the authors measured absorbance to test cell viability by a luminescence-based viability assay (CellTiter-Glo). Please correct the formula and text in the relevant method section.

Author Response

We sincerely thank you for your positive approach. The correction that should have been made in the last revision was overlooked. Thank you for your warning. The formula in the relevant method section has been removed and the text has been corrected.

(The section ‘2.3 Determination of in vitro antileishmanial activity’ has been revised again. The formulation previously included under this section has been removed. Relevant references are already provided within the Methods section. As you also pointed out, in the method described under the section titled ‘2.3 Determination of in vitro antileishmanial activity’, cell viability was assessed using the CellTiter-Glo® Luminescent Cell Viability Assay kit. The luminescence signals obtained through this method were measured as Relative Light Units (RLU), and cell viability percentages were calculated accordingly.)

Kind regards.

Round 3

Reviewer 2 Report

Comments and Suggestions for Authors

The authors responses are not clear. 

Author Response

Dear reviewer,
First of all, it is true that the test compounds in this study are not original compounds. The findings related to the syntheses have been previously published in a chemistry journal. We have clearly stated this information in the relevant section of the manuscript (you can see it in the revised file).  On the other hand, the antilesmanial effects of the tested compounds were demonstrated for the first time in this study and these findings are still important in the context of biological activity. In addition, an in silico approach was also presented in this study regarding the mechanisms of the antilesmanial effects of the test substances. I
Finally, regarding the in-silico studies, we made extensive revisions based on the comments of other reviewers. 
Thank you for your valuable contributions.
Kind regards.